# The Molecular Mechanism by Which miR-211-5p Regulates the Proliferation and Differentiation of Preadipocytes in Meat Rabbits by Targeting TPK1

**DOI:** 10.3390/ani15101497

**Published:** 2025-05-21

**Authors:** Xiaoxiao Zhang, Meigui Wang, Tao Tang, Jing Zhou, Wenqiang Sun, Xianbo Jia, Jie Wang, Hengwei Yu, Songjia Lai

**Affiliations:** College of Animal Science and Technology, Sichuan Agricultural University, Chengdu 611130, China; zxx18137928846@163.com (X.Z.); wmg1987797495@163.com (M.W.); m18483220592@163.com (T.T.); 13669919401@163.com (J.Z.); wqsun2021@163.com (W.S.); jaxb369@sicau.edu.cn (X.J.); wjie68@163.com (J.W.); 18792427097@163.com (H.Y.)

**Keywords:** domestic rabbit, preadipocyte, miR-211-5p, TPK1

## Abstract

Rabbit meat, characterized by its high protein, low fat, and low cholesterol content, is regarded as a representative of healthy meats. Excessive fat deposition not only affects the flavor of the meat but also hinders the process of genetic improvement. Preadipocytes are specialized cells with the ability to proliferate and differentiate into adipocytes. Their proliferation and differentiation can affect fat deposition, thereby influencing the meat quality of livestock. Therefore, studying the molecular mechanisms underlying these processes is of great significance for both the livestock industry and the meat production sector. Our research findings indicate that microRNAs can inhibit the proliferation of rabbit preadipocytes and promote their differentiation by suppressing gene expression. This reveals a novel molecular mechanism by which microRNAs regulate adipogenesis, providing new insights into lipid deposition in livestock.

## 1. Introduction

Adipose tissue regulates lipid storage, intramuscular fat deposition, and flavor compound formation, playing a vital role in livestock health and meat quality [1,2]. Preadipocyte differentiation, regulated by microenvironmental signals and transcription factors, involves a complex cascade of events leading to adipocyte maturation and functional specialization [3]. MicroRNAs (miRNAs) belong to a highly conserved family of short non-coding RNAs that function as critical post-transcriptional regulators. Through the RNA interference pathway, miRNAs target and modulate the expression of downstream genes at the post-transcriptional level, thereby playing a pivotal role in gene expression regulation [4]. The processes of adipocyte proliferation and differentiation are intricately regulated by a variety of microRNAs (miRNAs), which exert both positive and negative regulatory effects. These miRNAs modulate key signaling pathways and gene networks involved in adipogenesis, thereby playing critical roles in lipid metabolism and fat deposition. In studies on lipid deposition in sheep, miR-136 was found to promote the proliferation of adipose-derived stromal vascular fractions (SVFs) while attenuating their adipogenic differentiation [5]. Overexpression of miR-425-5p inhibited porcine intramuscular adipogenic differentiation, downregulating PPARγ, FABP4, and FASN, while its inhibition promoted adipogenesis [6]. As a negative regulator of STAT3, miR-125a-5p may simultaneously exert dual functions of promoting proliferation and inhibiting differentiation in 3T3-L1 preadipocytes [7]. miR-130b can significantly inhibit subcutaneous fat deposition in pigs. During the early stages of differentiation, overexpression of miR-130b markedly reduces lipid accumulation by suppressing cell proliferation and inducing apoptosis [8]. Numerous miRNAs and their target genes involved in the regulation of adipocyte differentiation have been identified, leading to the gradual establishment of a comprehensive regulatory network. This network highlights the critical roles of miRNAs in modulating adipogenesis and lipid metabolism.

Studies demonstrate that miR-211-5p regulates key biological processes, including tumor microenvironment immune responses, inflammatory signaling pathways, cell proliferation, apoptosis, migration, invasion, and cell cycle progression, by targeting specific mRNA molecules [9,10]. In glioma cells, miR-211-5p regulates HOXC8 expression, thereby modulating cellular proliferation and migration [11]. In the context of lipid metabolism regulation, our previous study utilized transcriptomic sequencing to compare rabbits fed a high-fat diet (HFD) with those on a standard normal diet. Differential expression analysis identified miR-211-5p as significantly upregulated in the HFD group (*p* < 0.05, log2FC = 2.3), suggesting its potential role in mediating diet-induced metabolic adaptations [12]. Research has shown that miR-211 negatively regulates ELOVL6, a crucial enzyme involved in fatty acid synthesis and lipid metabolism, indicating its role as a key regulator of lipid metabolism and adipogenesis in bovine adipose tissue [13]. Building on these findings implicating miR-211-5p in lipid metabolism, we aimed to systematically characterize its functional role in adipocyte proliferation and differentiation.

To address the unresolved mechanistic aspects, we performed gain-of-function (miR-211-5p mimic transfection) and loss-of-function (antagomir-mediated inhibition) experiments in rabbit preadipocytes. This experimental framework enabled a comprehensive assessment of miR-211-5p’s effects on both proliferative capacity and differentiation potential.

## 2. Results

### 2.1. miR-211-5p Inhibits the Proliferation of Preadipocytes

Transfection efficiency was validated by comparative analysis of miR-211-5p expression levels across experimental groups. The mimic-transfected group exhibited a statistically significant upregulation in miR-211-5p relative expression compared to the negative control (NC) group (*p* < 0.001), confirming successful implementation of miR-211-5p overexpression (Figure 1A). Conversely, the inhibitor-treated group demonstrated a marked reduction in miR-211-5p expression levels relative to the inhibitor negative control (INC) group (*p* < 0.001), verifying effective suppression of endogenous miR-211-5p through specific inhibitor transfection (Figure 1B). EdU assay results revealed that, compared to the negative control (NC) group, the miR-211-5p mimic-transfected group exhibited a statistically significant reduction in proliferation rates. miR-211-5p inhibitor treatment resulted in markedly enhanced proliferative activity relative to the inhibitor-specific negative control (INC) group (Figure 1C–E). Quantitative real-time polymerase chain reaction (RT-qPCR) results indicated that, compared to the NC group, the expression of CDK4 and PCNA was downregulated in the miR-211-5p mimic group, whereas their expression was upregulated in the miR-211-5p inhibitor group (Figure 1F,G). The results of the CCK-8 assay showed that at 0 h, 24 h, 48 h, and 72 h after transfection, the absorbance at 450 nm in the miR-211-5p mimic group was significantly decreased, while the miR-211-5p inhibitor group showed the opposite trend (Figure 1H,I). These results collectively suggest that miR-211-5p exerts an inhibitory effect on the proliferation of rabbit preadipocytes.

### 2.2. miR-211-5p Promotes the Differentiation of Rabbit Preadipocytes

To explore the effect of miR-211-5p on the differentiation of rabbit preadipocytes, we established a differentiation model of rabbit preadipocytes. Oil Red O staining was performed at 0 d, 2 d, 4 d, 6 d, and 8 d, respectively. The results showed that with the increase of time, adipocytes differentiated rapidly, and lipid droplets increased. This result was also confirmed by measuring the absorbance at 510 nm using a microplate reader (Figure 2A,B). RT-qPCR was used to detect the gene expression levels of C/EBPα and PPARγ at different time points, and it was found that their expression levels showed a gradually increasing trend (Figure 2C,D). We further investigated the expression pattern of miR-211-5p during the differentiation process of rabbit preadipocytes. The results showed that the expression level of miR-211-5p increased significantly with the increase of days (*p* < 0.001), suggesting that miR-211-5p may play a potential regulatory role in the differentiation of rabbit preadipocytes (Figure 2E). These research results collectively indicate that the differentiation model of rabbit preadipocytes has been successfully established.

After 4 days of transfection, Oil Red O staining under microscopic observation revealed that lipid accumulation in the mimic group was higher than that in the NC group, while the inhibitor group showed lower lipid accumulation compared to the INC group. Quantitative analysis at 510 nm absorbance confirmed these findings (Figure 2F–H). RT-qPCR results demonstrated that the expression levels of PPARγ and FABP4 were significantly upregulated in the mimic group, whereas these genes were downregulated in the inhibitor group (Figure 2I,J). WB analysis showed that, compared to the NC group, PPARγ expression was increased in the mimic group and decreased in the inhibitor group (Figure 2K,L).

These results suggest that miR-211-5p may play a positive regulatory role in the differentiation of rabbit preadipocytes.

### 2.3. Analysis of Sequencing Results

To investigate the effects of miR-211-5p overexpression on gene expression profiles during the proliferation and differentiation of rabbit preadipocytes, we transfected cells with miR-211-5p mimic and miR-211-5p negative control (NC), followed by a 2-day induction period. Subsequently, the cell samples were sent to Baimaike Biotechnology Co., Ltd. (Beijing, China) for transcriptome sequencing. Using rabbit preadipocytes as the experimental material, we constructed six cDNA libraries (*n* = 3) for high-throughput RNA sequencing analysis. All samples yielded Clean Data exceeding 5.80 Gb, with Q30 base percentages reaching 94.05% or higher, indicating high-quality libraries suitable for subsequent bioinformatics analysis. A total of 3335 novel genes were identified, among which 862 were functionally annotated. Differential gene expression analysis was performed using a threshold of |Fold Change| ≥ 1.5 and *p*-value < 0.05 as screening criteria. Utilizing the DESeq2 algorithm, we identified 147 differentially expressed genes (DEGs) from the sequencing libraries, comprising 100 upregulated genes and 47 downregulated genes. From the sequencing results, three upregulated and three downregulated genes were randomly selected for validation. The results of RT-qPCR showed that the expression patterns of genes such as HSP90AA1, ELN, FKBP10, HEXB, SELENOP, and IGFBP2 were consistent with the sequencing data (Figure 3A). A volcano plot was constructed using log2(fold change) and -log10(*p*-value) values to visualize differentially expressed genes (Figure 3B). Gene Ontology (GO) analysis revealed that the differentially expressed genes were primarily involved in biological processes such as biological adhesion, regulation of transcription, molecular function regulation and transducer activity, cell proliferation, cellular response to stimuli, and multicellular organismal processes (Figure 3C). Kyoto Encyclopedia of Genes and Genomes (KEGG) pathway analysis of the differentially expressed genes showed significant enrichment in pathways including ketone body synthesis and degradation, parathyroid hormone synthesis, secretion and action, valine, leucine, and isoleucine biosynthesis, apoptosis, MAPK signaling pathway, cytosolic DNA-sensing pathway, FoxO signaling pathway, and Wnt signaling pathway. The enrichment results were visualized using bar graphs, displaying the top 20 pathways with the most significant *q*-values (Figure 3D). Furthermore, utilizing the ENCORI database, we identified 903 candidate genes containing potential binding sites for miR-211-5p. By intersecting these candidates with our sequencing results, we obtained 81 potential target genes (Figure 3E).

### 2.4. The Role of TPK1 in Regulating the Proliferation and Differentiation of Rabbit Preadipocytes

To further explore the potential mechanisms by which miR-211-5p regulates adipogenesis, we utilized the ENCORI online database to screen for candidate genes containing potential binding sites for miR-211-5p. From the 903 candidate genes identified, we intersected them with our sequencing results and obtained 81 potential target genes (Figure 1E). The analysis revealed that *TPK1* is a target gene of miR-211-5p. RT-qPCR results showed that *TPK1* expression was significantly downregulated in the miR-211-5p mimic group compared to the control group, whereas it was upregulated in the inhibitor group (Figure 4A). The study constructed wild-type (wt-TPK1) and mutant (mut-TPK1) plasmids of TPK1. Luciferase reporter assays demonstrated that the luciferase activity of the group containing the wild-type *TPK1* mRNA 3′UTR was significantly suppressed, while no notable change was observed in the mutant group (Figure 4B,C). This indicates that *TPK1* is a direct target of miR-211-5p. To investigate the role of *TPK1* in the proliferation of rabbit preadipocytes, cells were transfected with si-TPK1 and si-TPK1-NC, and assays were performed after 2 days. The interference efficiency of *TPK1* was evaluated during the proliferation process (Figure 4D). RT-qPCR revealed that the expression levels of PCNA and CDK4 were increased in the si-TPK1-NC group but decreased in the si-TPK1 group (Figure 4E,F). CCK-8 assays indicated that the absorbance in the si-TPK1-NC group was significantly higher than that in the si-TPK1 group (Figure 4G). WB analysis showed that the protein levels of PCNA and CDK4 were elevated in the si-TPK1-NC group but reduced in the si-TPK1 group (Figure 4H–J). These results indicate that the proliferation of rabbit preadipocytes is inhibited after transfection with si-TPK1.

In terms of differentiation, after transfecting si-TPK1 and si-TPK1-NC and inducing differentiation for 4 days, we initially evaluated the interference efficiency of *TPK1*. The results demonstrated significant differences, indicating its suitability for subsequent experimental procedures (Figure 4K). RT-qPCR analysis showed that the expression levels of FABP4 and PPARγ were significantly higher in the si-TPK1 group compared to the si-TPK1-NC group (Figure 4L,M). Oil Red O staining showed that lipid accumulation in the si-TPK1 group was higher than that in the si-TPK1-NC group, and the measurement results at 510 nm absorbance were consistent with these observations (Figure 4N,O). WB results demonstrated that the protein levels of PPARγ and C/EBPα were increased in the si-TPK1 group (Figure 4P–R). These findings suggest that si-TPK1 plays a positive regulatory role in the differentiation of rabbit preadipocytes.

### 2.5. miR-211-5p Regulates the Proliferation and Differentiation of Rabbit Preadipocytes by Targeting TPK1

Finally, we investigated whether miR-211-5p regulates the proliferation and differentiation of rabbit preadipocytes by targeting *TPK1*. After co-transfection, the EdU proliferation assay revealed that the cell proliferation rate in the miR-211-5p inhibitor/si-TPK1 group was significantly lower than that in the miR-211-5p inhibitor/si-TPK1-NC group but significantly higher than that in the miR-211-5p INC/si-TPK1 group (Figure 5A,B). WB results showed that the protein expression levels of proliferation-related genes CDK4 and PCNA followed the same trend (Figure 5C–E). CCK-8 assay results at 0 h, 24 h, 48 h, and 72 h post-transfection were consistent with the EdU assay findings (Figure 5F). RT-qPCR results indicated that the mRNA expression level of CDK4 followed the same trend (Figure 5G).

Following co-transfection, cells were treated with differentiation induction medium and cultured for 4 days. WB analysis revealed that the protein expression levels of FABP4 and PPARγ were significantly higher in the miR-211-5p inhibitor/si-TPK1 group compared to the miR-211-5p inhibitor/si-TPK1-NC group but slightly lower than those in the miR-211-5p INC/si-TPK1 group, with significant differences observed (*p* < 0.01) (Figure 5H–J). RT-qPCR results indicated that the mRNA expression level of C/EBP*α* followed the same trend (Figure 5K). Oil Red O staining showed that the number of lipid droplets and absorbance in the miR-211-5p inhibitor/si-TPK1 group were significantly higher than those in the miR-211-5p inhibitor/si-TPK1-NC group but slightly lower than those in the miR-211-5p INC/si-TPK1 group, with significant differences observed (*p* < 0.01) (Figure 5L,M).

miR-211-5p inhibits the proliferation of rabbit preadipocytes and promotes their differentiation by targeting and suppressing the expression of the TPK1 gene.

## 3. Discussion

MicroRNAs (miRNAs) play critical roles in regulating intramuscular fat deposition and adipogenesis across species. In pigs, miR-429 and miR-425-5p target adipogenic genes *KLF9* and *KLF13* to modulate fat deposition [6,14], while miR-195 suppresses lipid accumulation during bovine preadipocyte differentiation [15]. In rabbits, miR-29b-3p promotes preadipocyte differentiation, and miR-9-5p upregulates key adipogenic markers (PPARγ, C/EBPα, FABP4) [16,17]. Additionally, miR-100 regulates lipid metabolism and marbling traits in meat-producing animals, whereas miR-34a and miR-143 influence adipogenesis by targeting *PPAR*γ and *IGF2R*, respectively [18]. These findings highlight miRNAs’ dual roles in physiological homeostasis and metabolic disorders, providing a foundation for RNA-based therapies targeting lipid-related diseases [19,20,21].

miR-211-5p, a recently identified cancer-associated miRNA, is frequently downregulated in various malignant tumors, where it functions as a tumor suppressor by targeting specific mRNAs [22]. Research has demonstrated its involvement in diverse biological processes, including immune modulation in the tumor microenvironment, regulation of inflammation-related signaling pathways, and control of tumor cell proliferation, apoptosis, migration, and invasion [23,24]. For instance, miR-211-5p suppresses hepatocellular carcinoma progression by negatively regulating acyl-CoA long-chain family member 4 (*ACSL4*) [25]. Additionally, its downregulation has been observed in renal cell carcinoma, cervical cancer, and glioma, where it contributes to tumor cell hyperproliferation [11,13,26]. These multifaceted roles underscore miR-211-5p’s significance as a key regulatory molecule in cancer biology. Previous studies on bovine adipose tissue formation revealed a negative correlation between miR-211 and *ELOVL6*, a key enzyme in fatty acid synthesis and lipid metabolism. These findings suggest miR-211 as a novel regulator of lipid metabolism, potentially modulating *ELOVL6*-mediated pathways to influence adipogenesis and fat deposition in cattle [27]. In a study on lipid metabolism regulation, transcriptomic sequencing of rabbits fed a high-fat diet (HFD) versus a standard normal diet (SND) revealed significant upregulation of miR-211-5p in the HFD group (*p* < 0.05, log_2_FC = 2.3). These findings suggest miR-211-5p may mediate diet-induced metabolic adaptations, highlighting its potential role in lipid metabolism [12]. Downregulation of miR-211-5p promotes carboplatin resistance in human retinoblastoma Y79 cells [28]. hsa_circ_0008285 facilitates the progression of cervical cancer by targeting miR-211-5p/SOX4 axis [29]. CircPAG1 interacts with miR-211-5p to promote the expression of E2F3 and inhibit high-glucose-induced apoptosis and oxidative stress in diabetic cataract [30]. miR-211-5p may play an inhibitory role in hepatocellular carcinoma (HCC) by suppressing the expression of ZEB2 [31]. To investigate the molecular mechanisms underlying lipid deposition, we examined the role of miR-211-5p in rabbit preadipocytes. Our findings reveal that miR-211-5p exhibits a dual regulatory function: it inhibits preadipocyte proliferation while promoting differentiation. These results suggest that miR-211-5p plays a critical role in adipogenesis by modulating both cellular expansion and maturation processes. This study provides new insights into miRNA-mediated regulation of adipose tissue development and highlights miR-211-5p as a potential target for managing lipid metabolism in livestock.

The identification of miRNA target genes involves two primary strategies: computational prediction and experimental validation. Computational approaches utilize tools like TargetScan and miRDB to analyze sequence complementarity between miRNAs and 3′UTR regions of target genes, along with binding energy calculations. While efficient, these predictions may exhibit algorithmic biases. Experimental validation employs qPCR and dual-luciferase reporter assays to confirm interactions. In our study, transcriptome sequencing data integrated with computational screening revealed a potential targeting relationship between miR-211-5p and *TPK1*, highlighting the importance of combining bioinformatic analysis with experimental verification to ensure robust findings in miRNA research.

*TPK1* (thiamine pyrophosphokinase 1) is a critical enzyme involved in thiamine metabolism, catalyzing the conversion of thiamine to its active form, thiamine pyrophosphate (*TPP*). *TPP* serves as an essential cofactor for key metabolic enzymes, including pyruvate dehydrogenase (*PDH*) and α-ketoglutarate dehydrogenase, which are pivotal in glucose and lipid metabolism. Recent studies have identified *TPK1* as a key player in lipid synthesis, with emerging evidence suggesting its broader role as a central regulator of glucose and lipid metabolism [32]. Additionally, *TPK1* modulates cellular maturation, highlighting its multifaceted involvement in metabolic and developmental processes [33]. *TPK1* enhances pyruvate dehydrogenase (*PDH*) activity by modulating thiamine metabolism, thereby promoting the differentiation of neuroblastoma cells. This mechanism highlights TPK1′s role in linking metabolic regulation to cellular development [34]. Our study revealed that miR-211-5p may regulate preadipocyte proliferation and differentiation in rabbits by targeting *TPK1*, consistent with previous findings. These results underscore TPK1′s pleiotropic role in coordinating metabolic regulation and developmental processes, positioning it as a critical node in energy homeostasis and cellular maturation. Further exploration of *TPK1*-mediated molecular mechanisms could advance our understanding of metabolic and developmental disorders, potentially identifying novel therapeutic targets through its dual regulatory functions.

## 4. Materials and Methods

### 4.1. Ethical Statement

All experimental procedures were conducted in strict compliance with the Institutional Animal Care and Use Committee (IACUC) guidelines of the College of Animal Science and Technology, Sichuan Agricultural University (Approval No. SYXK2019-187). Nine neonatal Tianfu Black rabbit offspring with confirmed health status and moderate body condition were randomly selected from a cohort of dams with synchronized pregnancies. Subsequently, the neonatal rabbits were euthanized in the Animal Research Laboratory of Sichuan Agricultural University following internationally recognized humane procedures to ensure animal welfare.

### 4.2. Animal Cell Collection and Culture

Under sterile conditions, adipose tissue surrounding the kidneys of newborn Tianfu black rabbits was collected. The tissue was enzymatically digested using Type I collagenase to isolate primary preadipocytes from the rabbits. The preadipocytes were then cultured in an incubator at 37 °C with 5% CO_2_, and the culture medium was replaced every 2–3 days. When the cells reached 70–80% confluence, they were passaged and cryopreserved for future use. Growth Medium: DMEM/F12 (Gibco, 11330032, Thermo Fisher Scientific, San Jose, CA, USA) supplemented with 10% fetal bovine serum (FBS, Gibco, Carlsbad, CA, USA). Induction Differentiation Medium: DMEM/F12 (Gibco, 11330032) containing 5% fetal bovine serum (FBS, Gibco, Carlsbad, CA, USA), 1 µM dexamethasone, 3-isobutyl-1-methylxanthine (IBMX), and 10 µg/mL insulin (Solarbio, Beijing, China). Maintenance Medium: DMEM/F12 (Gibco, 11330032) containing 5% fetal bovine serum (FBS, Gibco, Carlsbad, CA, USA) and 10 µg/mL insulin (Solarbio, Beijing, China).

### 4.3. Bioinformatic Analysis of Sequencing Data

RNA samples were prepared and subjected to stringent quality control. Initial quantification was performed using a Qubit 3.0 Fluorometer, ensuring a concentration of at least 1 ng/μL. Subsequently, the insert size of the library was assessed using the Qsep400 High-Throughput Analysis System (Bioptic, New Taipei City, Taiwan). Once the insert size met the expected criteria, the effective concentration of the library (effective library concentration >2 nM) was accurately quantified using RT-qPCR. After passing the library quality check, different libraries were pooled based on their effective concentrations and the required off-target data volume, followed by sequencing on the Illumina platform (San Diego, CA, USA). Differential expression analysis was conducted using DESeq2 with a significance threshold of *p* ≤ 0.05.

### 4.4. RNA Extraction and Quantitative Real-Time PCR (RT-qPCR)

Total RNA was extracted from the samples using a Total RNA Extraction Kit (Solarbio, Beijing, China) following the manufacturer’s guidelines. The integrity of the RNA was assessed by 1.0% agarose gel electrophoresis. First-strand cDNA synthesis for total RNA and small RNA was performed using the PrimeScript RT Reagent Kit (Takara, Beijing, China) and the SYBR^®®^ PrimeScript™ miRNA RT Kit (Takara, Beijing, China), respectively, according to the instructions provided with each kit. qPCR was carried out in a 10 µL reaction volume using GAPDH and U6 as housekeeping genes. The reverse primer for miR-211-5p and the primers for U6 were provided in the respective kits. Subsequently, an amplification reaction was carried out using the CFX96 real-time PCR system (Bio-Rad, Hercules, CA, USA).

### 4.5. Cell Induced Differentiation

Preadipocytes were maintained in growth medium supplemented with 10% fetal bovine serum (FBS, Gibco, Carlsbad, CA, USA) and incubated in a humidified incubator (Thermo Fisher Scientific, San Jose, CA, USA) at 37 °C with 5% CO_2_. The medium was replaced every 24 h. When the cells reached 70–80% confluence, they were induced to differentiate using differentiation medium containing 5% fetal bovine serum (FBS, Carlsbad, CA, USA), 1 µmol/L dexamethasone, 0.5 mmol/L 3-isobutyl-1-methylxanthine (IBMX), and 10 µg/mL insulin (Solarbio, Beijing, China) for 2 days. Subsequently, the medium was replaced with maintenance medium containing 5% fetal bovine serum and 10 µg/mL insulin for an additional 2 days. Finally, the medium was replaced with growth medium (GM) to obtain mature adipocytes.

### 4.6. Transfection

miR-211-5p mimic, miR-211-5p inhibitor, miR-211-5p-NC, miR-211-5p-INC, si-TPK1-NC, and si-TPK1 were transfected using LipoMax (Sudgen, Nanjing, China). In the experiment, when the cell density reached 80%, transfection was performed with LipoMax. After 6 h, the medium was replaced with either induction medium or regular medium. RNA oligonucleotides were synthesized by Sangon Biotech (Shanghai, China). The miRNA mimics are chemically synthesized mature double-stranded miRNAs designed to enhance the function of endogenous miRNAs. These include a sequence identical to the mature miRNA of interest and a complementary sequence to the mature miRNA. The miRNA inhibitors are chemically synthesized single-stranded miRNAs with methoxy modifications, specifically designed to target and inhibit the activity of endogenous miRNAs in cells, enabling loss-of-function studies of miRNAs. NC and INC serve as negative controls.

### 4.7. Western Blotting (WB)

Western blot (WB) is a widely used experimental technique in the field of molecular biology, primarily employed to detect the expression levels and molecular weights of specific proteins in samples. Total cellular proteins were extracted using a commercial protein extraction kit (Solarbio, Beijing, China) following the manufacturer’s instructions. Protein concentration was determined using a Bradford protein assay kit (Novoprotein, Shanghai, China). Subsequently, the proteins were separated by SDS-PAGE electrophoresis and transferred onto a PVDF membrane. PVDF membranes were subjected to blocking with a 5% skim milk solution to mitigate nonspecific binding. Following blocking, the membrane was incubated with the corresponding primary antibody at 4 °C for 8 h, and then with a secondary antibody (Goat Anti-Rabbit IgG H&L (HRP), Zen Bioscience, Chengdu, China) for 2 h. The membrane was washed three times with TBST solution containing 0.1% Tween-20. The PVDF membrane was then placed on the stage of the chemiluminescence imaging system (Servicebio SCG-W3000, Wuhan, China) in a light-shielded environment. Then, an appropriate amount of ECL chemiluminescence reagent (Beyotime, Shanghai, China) was added to the membrane. Subsequently, the exposure time was set, and the image was captured. Quantitative analysis of the gray values of the bands was performed using ImageJ (1.8.0) software, with β-actin as the reference protein. Primary antibodies against PCNA and CDK4 were purchased from Zen Bioscience (Chengdu, China). Primary antibodies against PPARγ, C/EBPα, β-actin, and FABP4 were purchased from ABclonal (Wuhan, China).

### 4.8. Cell Counting Kit 8 (CCK-8) Assay

Preadipocytes were seeded into 96-well plates and transfected when the cell density reached approximately 70%. According to the manufacturer’s instructions, at 0, 24, 48, and 72 h post-transfection, 10 µL of CCK-8 reagent was added to each well, followed by incubation for two hours. The absorbance at 450 nm (OD 450 nm) was measured using a Thermo Scientific™ Varioskan LUX (Thermo Scientific, Waltham, MA, USA). The absorbance data were plotted and analyzed using GraphPad Prism 9 (GraphPad Software Inc., La Jolla, CA, USA).

### 4.9. EdU Proliferation Assay

Preadipocytes were seeded into 24-well plates. Forty-eight hours post-transfection, the cells were incubated in complete medium containing 50 µL of 5-ethynyl-2′-deoxyuridine (EdU, RiboBio, Guangzhou, China) for 2 h, followed by immunostaining. Fluorescence microscopy was used to capture at least five images per group. The Image-Pro Plus 6.0 software (Media Cybernetics, Inc., Rockville, MD, USA) was used for cell counting. The proliferation rate of preadipocytes in rabbits was determined by the ratio of proliferating cells (red) to total cells (blue). The unit is percentage.

### 4.10. Oil Red O Staining

Preadipocytes were seeded into 35 mm cell culture dishes (NEST Biotechnology, Wuxi, China). Four days after transfection, the cells were stained. Prior to staining, the cells were washed twice with PBS and then fixed with 10% paraformaldehyde for 30 min. Oil Red O staining solution was prepared according to the manufacturer’s instructions and used to stain the cells. Images were captured using an inverted microscope.

Subsequently, the cells were washed with PBS several times to remove the excess dye. After no precipitate was observed, 2 mL of 100% isopropanol was added to the cell culture dish to extract the intracellular lipids. At this time, we carefully covered the lid tightly. Ten minutes later, the solution was transferred to a 96-well plate, and the absorbance was promptly measured using the Thermo Fisher Scientific Varioskan LUX microplate reader at 510 nm (OD 510 nm).

### 4.11. Dual-Luciferase Reporter Assay

We predicted the target genes of miR-211-5p based on sequence homology by integrating data from the ENCORI database and transcriptome sequencing results. TPK1 was selected as a potential target gene of miR-211-5p. The luciferase reporter plasmids, including the wild-type (WT) and mutant (MUT) constructs, were synthesized by Tsingke Biotechnology Co., Ltd. Subsequently, 293T cells were seeded into 24-well plates (NEST Biotechnology, Wuxi, China). When the cell density reached 70%, miR-211-5p mimic or negative control (NC) was co-transfected with either the WT or MUT plasmid using LipoMax (Sudgen, Nanjing, China). After 24 h, luciferase activity was measured using the Duo-Lite™ Luciferase Assay System (Vazyme, Nanjing, China).

### 4.12. Statistical Analysis

All data were analyzed using SPSS 20.0 statistical software. For data with a normal distribution, the results are expressed as the mean ± standard deviation (SD). In addition, one-way or two-way ANOVA was used for multiple comparisons. The significance levels are indicated by * *p* < 0.05,** *p* < 0.01, and *** *p* < 0.001, respectively.

## 5. Conclusions

Our research results indicate that miR-211-5p inhibits the proliferation of rabbit preadipocytes and promotes their differentiation, and there is a direct targeting relationship between miR-211-5p and TPK1. miR-211-5p inhibits the proliferation of rabbit preadipocytes and promotes their differentiation by targeting and suppressing the expression of the TPK1 gene. This reveals a novel molecular mechanism by which miR-211-5p regulates adipogenesis, providing new insights into lipid deposition in livestock.

## Figures and Tables

**Figure 1 animals-15-01497-f001:**
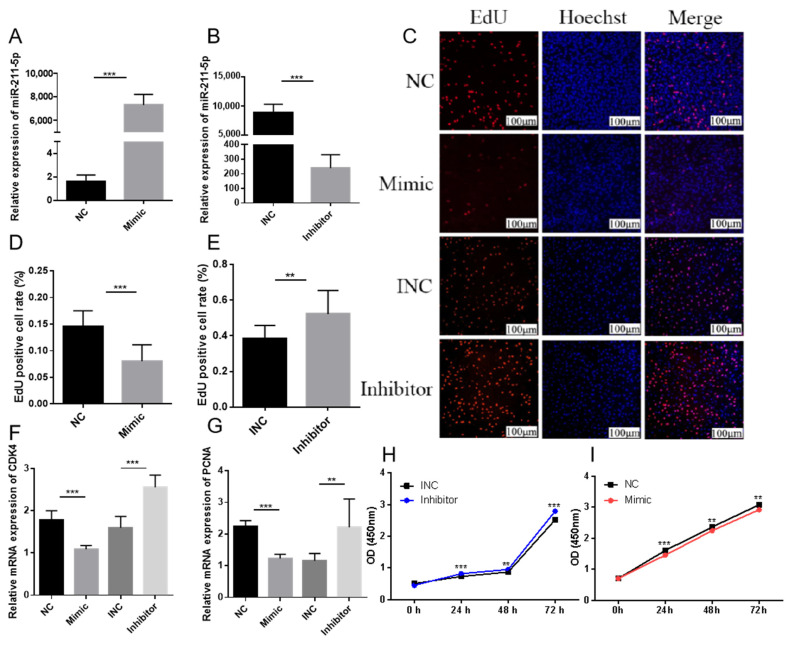
miR-211-5p inhibits the proliferation of rabbit preadipocytes. (**A**,**B**) Transfection efficiency of miR-211-5p mimic and inhibitor (*n* = 10). (**C**) EdU proliferation assay of preadipocytes transfected with miR-211-5p mimic, NC, miR-211-5p inhibitor, and INC. Proliferating cells (red), total cells (blue) (*n* = 10). (**D**,**E**) The proliferation rate of preadipocytes in rabbits. The proliferation rate is determined by the ratio of proliferating cells (red) to total cells (blue). (**F**,**G**) Expression levels of PCNA and CDK4 (*n* = 9). (**H**,**I**) CCK-8 cell viability assay showing the absorbance of preadipocytes at 0 h, 24 h, 48 h, and 72 h. Data are presented as mean ± SD. ** *p* < 0.01, *** *p* < 0.001.

**Figure 2 animals-15-01497-f002:**
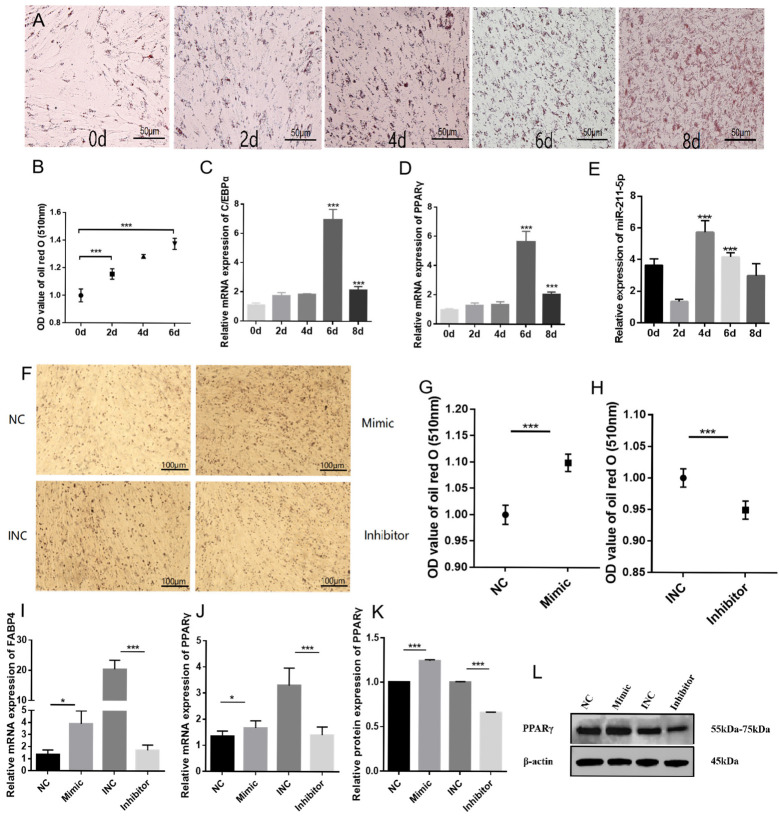
miR-211-5p promotes the differentiation of rabbit preadipocytes. (**A**) Oil Red O staining of lipid droplets at 0, 2, 4, 6, and 8 days post-differentiation. (**B**) Dissolve the Oil Red O staining agent with 100% isopropanol, and measure the OD value at 510 nm (*n* = 7). (**C**,**D**) mRNA expression levels of PPARγ and C/EBP*α* during the differentiation of rabbit preadipocytes (*n* = 6). (**E**) Relative expression levels of miR-211-5p during preadipocyte differentiation (*n* = 9). (**F**–**H**) Oil Red O staining of lipid droplets and measure the OD value at 510 nm (*n* = 7). (**I**,**J**) mRNA expression levels of PPARγ and FABP4 during the differentiation of rabbit preadipocytes transfected with NC, miR-211-5p mimic, INC, and miR-211-5p inhibitor (*n* = 8). (**K**,**L**) Determination of the gray value of protein expression level of PPARγ during the differentiation of preadipocytes in rabbits (*n* = 3). Data are presented as mean ± SD. * *p* < 0.05; *** *p* < 0.001.

**Figure 3 animals-15-01497-f003:**
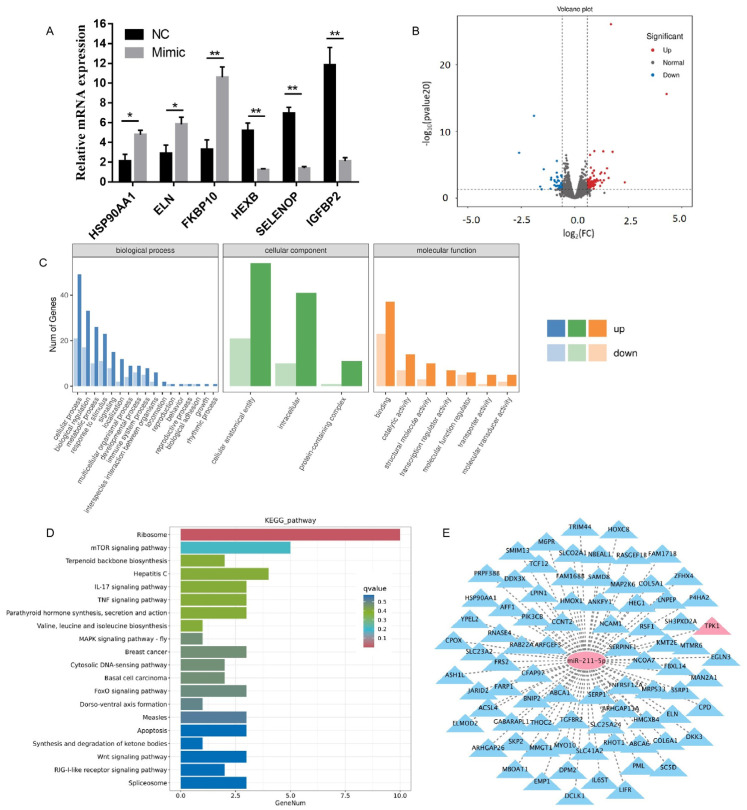
Analysis of differentially expressed genes. (**A**) Validation of gene expression from sequencing results. (**B**) Volcano plot of differentially expressed genes constructed based on log_2_(fold change) and -log_10_(*p*-value). (**C**) Gene Ontology (GO) analysis of differentially expressed genes. (**D**) Kyoto Encyclopedia of Genes and Genomes (KEGG) pathway analysis of differentially expressed genes. (**E**) Intersection of miR-211-5p target genes predicted by sequencing results and RNA hybridization software. Data are presented as mean ± SD. * *p* < 0.05; ** *p* < 0.01.

**Figure 4 animals-15-01497-f004:**
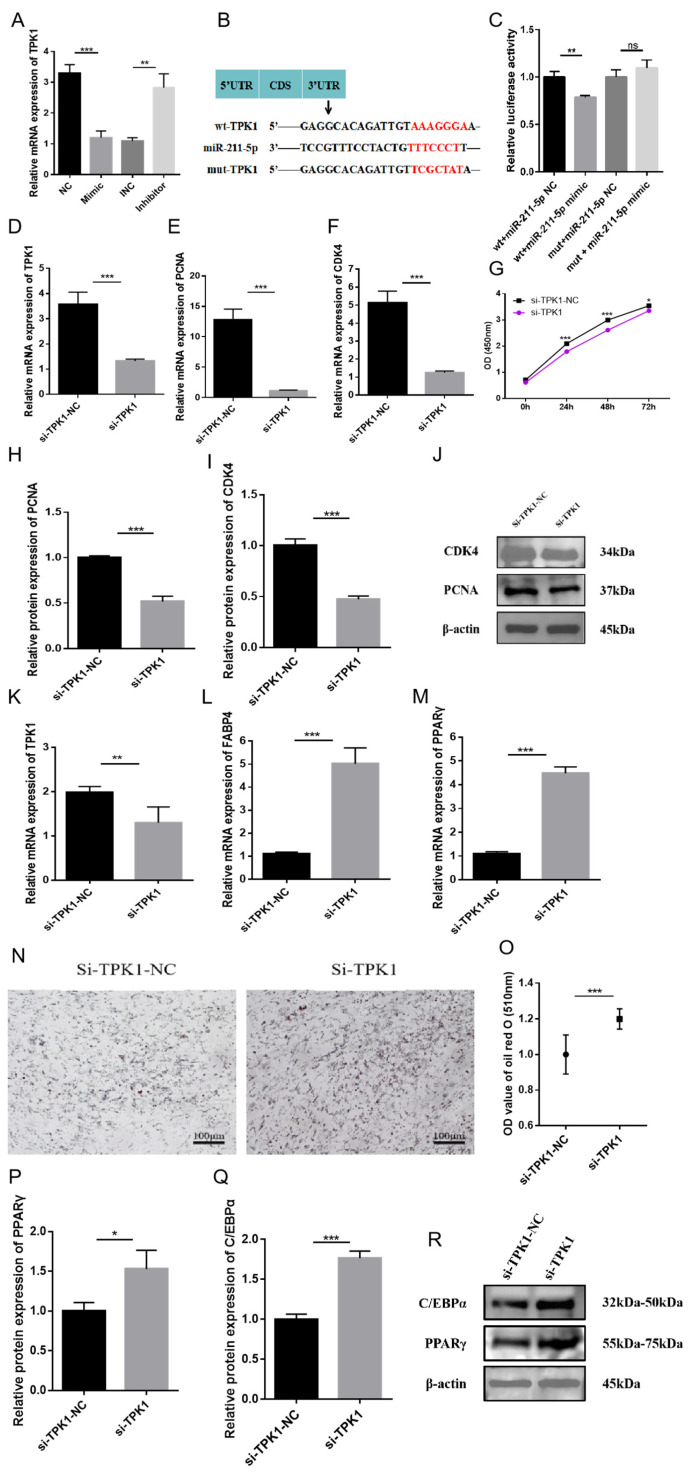
Effects of *TPK1* on the proliferation and differentiation of rabbit preadipocytes. (**A**) mRNA expression levels of *TPK1* in preadipocytes transfected with miR-211-5p mimic, NC, miR-211-5p inhibitor, and INC (*n* = 3). (**B**) Construct wild-type and mutant plasmids based on the binding sites between the TPK1 gene and miR-211-5p. (**C**) Luciferase assay performed by co-transfecting 293T cells with miR-211-5p mimic or NC and either wild-type or mutant TPK1 plasmids. (**D**) Interference efficiency of *TPK1* during the proliferation process. (**E**,**F**) mRNA expression levels of CDK4 and PCNA during the proliferation of preadipocytes transfected with si-TPK1 and si-TPK1-NC (*n* = 9). (**G**) CCK-8 cell viability assay showing the absorbance of preadipocytes at 0, 24, 48, and 72 h post-transfection with si-TPK1 and si-TPK1-NC (*n* = 7). (**H**–**J**) Protein levels of CDK4 and PCNA during the proliferation of preadipocytes transfected with si-TPK1 and si-TPK1-NC (*n* = 3). (**K**) Interference efficiency of *TPK1* during the differentiation process. (**L**,**M**) mRNA expression levels of FABP4, PPARγ during the proliferation of preadipocytes transfected with si-TPK1 and si-TPK1-NC (*n* = 3). (**N**) Oil Red O staining of lipid droplets. (**O**) Dissolve the Oil Red O staining agent with 100% isopropanol, and measure the OD value at 510 nm (*n* = 8). (**P**–**R**) Determination of protein levels and gray values of PPARγ and C/EBPα during the proliferation process of preadipocytes transfected with si-TPK1 and si-TPK1-NC (*n* = 3). Data are presented as mean ± SD. * *p* < 0.05; ** *p* < 0.01, *** *p* < 0.001.

**Figure 5 animals-15-01497-f005:**
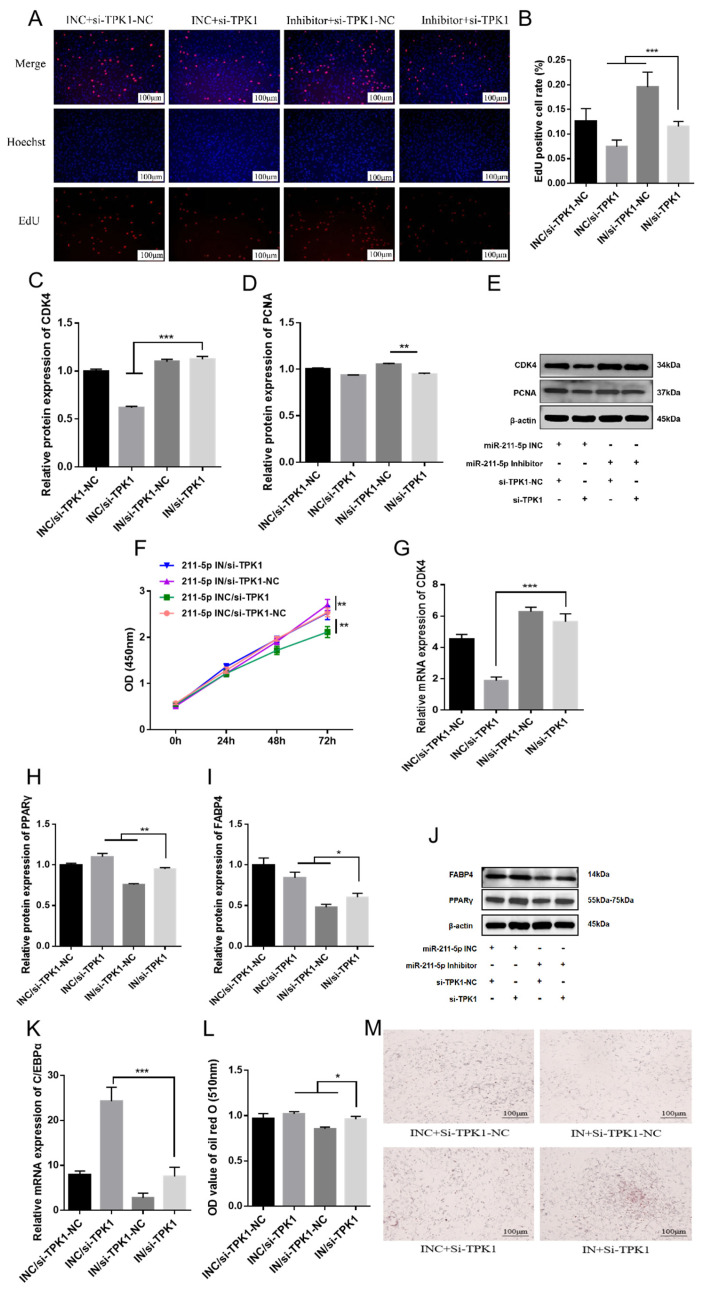
miR-211-5p regulates the proliferation and differentiation of rabbit preadipocytes by targeting *TPK1*. (**A**) EdU proliferation assay of preadipocytes transfected with miR-211-5p INC/si-TPK1-NC, miR-211-5p inhibitor/si-TPK1-NC, miR-211-5p INC/si-TPK1, and miR-211-5p inhibitor/si-TPK1 (*n* = 5). Red fluorescence indicates EdU-positive cells, and blue fluorescence represents Hoechst-stained nuclei. (**B**) The proliferation rate of preadipocytes in rabbits. The proliferation rate is determined by the ratio of proliferating cells (red) to total cells (blue). (**C**–**E**) Protein expression levels of PCNA and CDK4 in rabbit preadipocytes transfected (*n* = 3). (**F**) CCK-8 cell viability assay showing the absorbance of preadipocytes at 0, 24, 48, and 72 h post-transfection (*n* = 7). (**G**) mRNA expression levels of CDK4 in rabbit preadipocytes transfected. (**H**–**J**) Protein expression levels of FABP4 and PPARγ in rabbit preadipocytes transfected (*n* = 3). (**K**) mRNA expression levels of C/EBPα in rabbit preadipocytes transfected. (**L**) Oil Red O staining results of lipid droplets at 4 days post-transfection. (**M**) Dissolve the Oil Red O staining agent with 100% isopropanol, and measure the OD value at 510 nm (*n* = 5). Data are presented as mean ± SD. * *p* < 0.05; ** *p* < 0.01, *** *p* < 0.001.

## Data Availability

The datasets presented in this study can be found in online repositories.

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
