# Peer review of "The Molecular Mechanism by Which miR-211-5p Regulates the Proliferation and Differentiation of Preadipocytes in Meat Rabbits by Targeting TPK1"

_animals, 2025, doi:10.3390/ani15101497_

Round 1
Reviewer 1 Report
Comments and Suggestions for Authors To address the unresolved mechanistic aspects, the authors conducted 76 gain-of-function (miR-211-5p mimic transfection) and loss-of-function (antagomir-mediated inhibition) experiments in rabbit preadipocytes. This experimental framework enabled a comprehensive analysis of the effects of miR-211-5p on proliferative capacity and differentiation potential. I find this manuscript highly significant and interesting, and I recommend its acceptance with minor revisions. 1. Figure Clarity: The presentation of some figures (particularly Figures 4 and 5) appears overcrowded. It is recommended to optimize the layout by adjusting subfigure spacing and adding clear panel labels (A, B, C, etc.) with descriptive titles.
2. Gene Nomenclature Standardization: The gene symbol formatting in line 123 requires correction (e.g., "CEBPα" should be properly typeset using italics and Greek symbols: C/EBPα).
3. Language Precision: Several grammatical inconsistencies were identified, particularly regarding subject-verb agreement (e.g., the ambiguous subject in "miR-211-5p inhibitor/si-TPK1 group was significantly lower..." - specify whether this refers to protein levels, mRNA expression, or other metrics).
4. Literature Balance: While the reference to recent studies (2024) supporting TPK1's metabolic role is appropriate, key background statements about miR-211-5p's oncogenic functions would benefit from citation of foundational works (e.g., Seminal papers from 2010-2018 establishing miR-211-5p's cancer-related mechanisms).
5. Terminology Consistency: The abbreviation format for quantitative real-time PCR ("qRT-PCR" vs "RT-qPCR") should be standardized throughout the manuscript. Please maintain consistent usage in both main text and Methods section 4.4.
Reviewer 2 Report
Comments and Suggestions for Authors
This study completely and systematically confirmed miR-211-5p was shown to regulate the proliferation and differentiation of rabbit white preadipocytes by targeting TPK1. The experiment is very comprehensive and rigorous, and can be used as a model for the same kind of research. I have only a few comments to suggest the author consider:
- Please include species in the title.
- Please keep WB strips in the same color as possible.
- How conserved is miR-211-5p among the mentioned species such as cattle and rabbits?
Reviewer 3 Report
Comments and Suggestions for Authors
General comments:
The following review is for the manuscript “miR-211-5p regulates the proliferation and differentiation of preadipocytes by targeting TPK1”. This miRNA has been found to regulate lipid metabolism/deposition by altering the proliferation and differentiation of preadipocytes. Researchers present the case that this miR-211-5p does this by interacting with TPK1.
The manuscript does not follow formatting guidelines. The materials and methods should come after the introduction and before the results.
In general, the methods are not sufficiently described. This experiment could not be replicated with the information provided. I recommend the authors go through and add more detail. Some specific examples are provided in the line specific comments.
More detail is needed in the statistical analysis section of the methods. Without seeing the statistical approach, it is difficult to determine if the analysis is sufficient. Authors state on line 469: for normally distributed data,…”. This implies that some data collected was not normally distributed. How was normality assessed? Was a non-parametric test performed to analyze this non-normal data or was the data transformed so it would be normally distributed. Was a power analysis done to estimate sample size? What type of T test was done, paired or unpaired? These details are needed.
In general, data presentation needs to be improved, and the steps of data generation need to be elaborated on. For instance, time series data is presented but not described anywhere. I need much more information before I can assess the experimental design.
Line specific/Section comments.
Abstract: The abstract goes right form intro to results without discussing the experimental design. I recommend restructuring in a way to increase the readability of this abstract by introducing the methods before the results.
Introduction:
Ln 36: Remove the word enhancement as this is subjective.
Ln 68: Abbreviations need to be introduced prior to usage.
Results: Should be after the materials and methods
Ln 85-87: This needs to be clarified as I do not know what you mean here. Why are the words added in parentheses. It is confusing and unclear as to what this is saying.
Ln 88: Positive cell rate of what? Differentiation or proliferation?
Ln 111: There is no discussion of different time points being assessed in the methods. This MUST be added in.
Ln 113: This is methods and should be described in that section.
Materials and Methods:
Ln 371: How many rabbits were used? Were they all from the same litter? How long after birth were they harvested? Where they euthanized prior to tissue collection, if so how?
Ln 397: Can you be more specific about what media you used?
Ln 427: Surely you blocked in a 5% nonfat milk solution and not the “skimmed milk powder”. Please update.
Ln 430: Washed 3x with what? 0.1% TBST?
Ln 431: How was the chemiluminescence detected? How did you image the blots and then analyze the images? How is the data presented? Is it relative to a reference protein or as a % of ponceau.
Ln 433: What company in Wuhan, China were the antibodies purchased?
Ln 454: How were the images analyzed?
Ln 455-457: I am assuming you did add the 2 mL of Oil Red O to determine how much of the stain was retained by the preadipocytes. I am assuming that you are assuming that relative absorbance will be positively correlated with stain extracted from the cells and therefore lipid accumulation. I like this idea, but the details as to why you did this is lacking. As you read, I made a lot of assumptions that should be explicitly stated in the literature. Please add some more information as to the purpose of this step. This is new to me, and it would be helpful if you could add a citation for the validation of the step of dye extraction and quantification. Also, how did you prevent the isopropanol from evaporating from the microplate? Did you cover/seal the plate? If so, what kind of cover/seal was used?
Figure 1: You have what ** represents in the figure caption, what about ***? What is the units for the EDU rate? The caption says the percentage of positive cells, the Y axis of the graphs should reflect this. A rate is a change over time and I believe you are just showing how many are positive. The line graphs in panel E should have an X axis title such as Time (h). Also, the spaces between the value and the unit is uneven. If you add in the x axis then you can remove the unit from each numerical value listed on the axis. What is the y axis of the figures showing? Optical density? Why not just say absorbance. OD is not defined anywhere in this manuscript.
Figure 2: What treatment group are you showing with your oil red o images in panel A? This is not clear. Again, this time sequence is not described anywhere in this manuscript nor is the quantification of these images. What is panel B showing? Did you quantify number of lipid droplets were area?
Figure 4: The letter denotating panel G covering up some text on the x axis of D.
Reviewer 4 Report
Comments and Suggestions for Authors
The presented manuscript by a group of co-authors is about the miRNAs importance in livestock issues. Through the RNA interference pathway, miRNAs target and modulate the expression of downstream genes at the post-transcriptional level, thereby playing a pivotal role in gene expression regulation. In this manuscript, it is mentioned that miR-211-5p is dysregulated in multiple malignancies and implicated in tumor cell proliferation, apoptosis, inflammation, and neurological processes. In addition, according to the authors of the manuscript, miR-211 has been shown to negatively regulate ELOVL6, suggesting its involvement in lipid metabolism and adipogenesis in bovine adipose tissue. The authors of the manuscript, in the presented research work, have demonstrated that the overexpression of miR-211-5p significantly reduced the mRNA expression levels of CDK4 and PCNA (p < 0.01) and inhibition of miR-211-5p produced the opposite effect, promoting cell proliferation. In addition, to investigate the interaction between miR-211-5p and TPK1, a co-transfection experiment was conducted using four experimental groups. RT-qPCR, WB, Oil Red O staining, and CCK-8 assays revealed that TPK1 is a direct target of miR-211-5p. miR-211-5p was shown to regulate the proliferation and differentiation of rabbit white preadipocytes by targeting TPK1. It is well structured manuscript and the results are presented in a good way. The discussion is also a valuable one, which includes several citations.
I would recommend to the Editor in Chief of the Journal to accept the manuscript for publication in the presented for.
